# Open-vocabulary Object Detection via Vision and Language Knowledge Distillation

**Xiuye Gu[1], Tsung-Yi Lin[2], Weicheng Kuo[1], Yin Cui[1]**
[1]Google Research, [2]Nvidia[*]
{xiuyegu, weicheng, yincui}@google.com    tsungyil@nvidia.com

## Abstract

We aim at advancing open-vocabulary object detection, which detects objects described by arbitrary text inputs. The fundamental challenge is the availability of training data. It is costly to further scale up the number of classes contained in existing object detection datasets. To overcome this challenge, we propose **ViLD**, a training method via **Vi**sion and **L**anguage knowledge **D**istillation. Our method distills the knowledge from a pretrained open-vocabulary image classification model (teacher) into a two-stage detector (student). Specifically, we use the teacher model to encode category texts and image regions of object proposals. Then we train a student detector, whose region embeddings of detected boxes are aligned with the text and image embeddings inferred by the teacher. We benchmark on LVIS by holding out all rare categories as novel categories that are not seen during training. ViLD obtains 16.1 mask $AP_r$ with a ResNet-50 backbone, even outperforming the supervised counterpart by 3.8. When trained with a stronger teacher model ALIGN, ViLD achieves **26.3** $AP_r$. The model can directly transfer to other datasets without finetuning, achieving 72.2 $AP_{50}$ on PASCAL VOC, 36.6 AP on COCO and 11.8 AP on Objects365. On COCO, ViLD outperforms the previous state-of-the-art (Zareian et al., 2021) by 4.8 on novel AP and 11.4 on overall AP. Code and demo are open-sourced at https://github.com/tensorflow/tpu/tree/master/models/official/detection/projects/vild.

## 1 Introduction

Consider Fig. 1, can we design object detectors beyond recognizing only *base categories* (*e.g.*, toy) present in training labels and expand the vocabulary to detect *novel categories* (*e.g.*, toy elephant)? In this paper, we aim to train an *open-vocabulary* object detector that detects objects in any novel categories described by text inputs, using only detection annotations in base categories.

Existing object detection algorithms often learn to detect only the categories present in detection datasets. A common approach to increase the detection vocabulary is by collecting images with more labeled categories. The research community has recently collected new object detection datasets with large vocabularies (Gupta et al., 2019; Kuznetsova et al., 2020). LVIS (Gupta et al., 2019) is a milestone of these efforts by building a dataset with 1,203 categories. With such a rich vocabulary, it becomes quite challenging to collect enough training examples for all categories. By Zipf's law, object categories naturally follow a long-tailed distribution. To find sufficient training examples for rare categories, significantly more data is needed (Gupta et al., 2019), which makes it expensive to scale up detection vocabularies.

On the other hand, paired image-text data are abundant on the Internet. Recently, Radford et al. (2021) train a joint vision and language model using 400 million image-text pairs and demonstrate impressive results on directly transferring to over 30 datasets. The pretrained text encoder is the key to the zero-shot transfer ability to arbitrary text categories. Despite the great success on learning image-level representations, learning object-level representations for open-vocabulary detection is still challenging. In this work, we consider borrowing the knowledge from a pretrained open-vocabulary classification model to enable open-vocabulary detection.

---

[*]Work done while Xiuye was a Google AI Resident and Tsung-Yi was at Google.

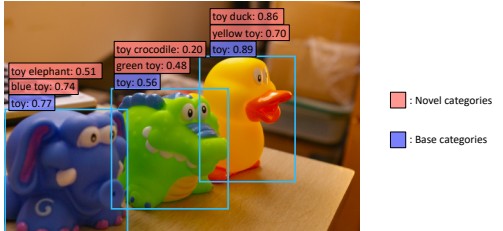

Figure 1: **An example of our open-vocabulary detector with arbitrary texts.** After training on base categories (purple), we can detect novel categories (pink) that are not present in the training data.

We begin with an R-CNN (Girshick et al., 2014) style approach. We turn open-vocabulary detection into two sub-problems: 1) generalized object proposal and 2) open-vocabulary image classification. We train a region proposal model using examples from the base categories. Then we use the pre-trained open-vocabulary image classification model to classify cropped object proposals, which can contain both base and novel categories. We benchmark on LVIS (Gupta et al., 2019) by holding out all rare categories as novel categories and treat others as base categories. To our surprise, the performance on the novel categories already surpasses its supervised counterpart. However, this approach is very slow for inference, because it feeds object proposals one-by-one into the classification model.

To address the above issue, we propose **ViLD** (**Vi**sion and **L**anguage knowledge **D**istillation) for training two-stage open-vocabulary detectors. ViLD consists of two components: learning with text embeddings (ViLD-text) and image embeddings (ViLD-image) inferred by an open-vocabulary image classification model, *e.g.*, CLIP. In **ViLD-text**, we obtain the text embeddings by feeding category names into the pretrained text encoder. Then the inferred text embeddings are used to classify detected regions. Similar approaches have been used in prior detection works (Bansal et al., 2018; Rahman et al., 2018; Zareian et al., 2021). We find text embeddings learned jointly with visual data can better encode the visual similarity between concepts, compared to text embeddings learned from a language corpus, *e.g.*, GloVe (Pennington et al., 2014). Using CLIP text embeddings achieves **10.1** $AP_r$ (AP of novel categories) on LVIS, significantly outperforming the **3.0** $AP_r$ of using GloVe. In **ViLD-image**, we obtain the image embeddings by feeding the object proposals into the pretrained image encoder. Then we train a Mask R-CNN whose region embeddings of detected boxes are aligned with these image embeddings. In contrast to ViLD-text, ViLD-image distills knowledge from both base and novel categories since the proposal network may detect regions containing novel objects, while ViLD-text only learns from base categories. Distillation enables ViLD to be *general* in choosing teacher and student architectures. ViLD is also *energy-efficient* as it works with off-the-shelf open-vocabulary image classifiers. We experiment with the CLIP and ALIGN (Jia et al., 2021) teacher models with different architectures (ViT and EfficientNet).

We show that ViLD achieves **16.1** AP for novel categories on LVIS, surpassing the supervised counterpart by **3.8**. We further use ALIGN as a stronger teacher model to push the performance to **26.3** novel AP, which is close (only 3.7 worse) to the 2020 LVIS Challenge winner (Tan et al., 2020) that is fully-supervised. We directly transfer ViLD trained on LVIS to other detection datasets without finetuning, and obtain strong performance of 72.2 $AP_{50}$ on PASCAL VOC, 36.6 AP on COCO and 11.8 AP on Objects365. We also outperform the previous state-of-the-art open-vocabulary detector on COCO (Zareian et al., 2021) by 4.8 novel AP and 11.4 overall AP.

## 2 RELATED WORK

**Increasing vocabulary in visual recognition:** Recognizing objects using a large vocabulary is a long-standing research problem in computer vision. One focus is zero-shot recognition, aiming at recognizing categories not present in the training set. Early works (Farhadi et al., 2009; Rohrbach et al., 2011; Jayaraman & Grauman, 2014) use visual attributes to create a binary codebook representing categories, which is used to transfer learned knowledge to unseen categories. In this direction, researchers have also explored class hierarchy, class similarity, and object parts as discriminative features to aid the knowledge transfer (Rohrbach et al., 2011; Akata et al., 2016; Zhao et al., 2017; Elhoseiny et al., 2017; Ji et al., 2018; Cacheux et al., 2019; Xie et al., 2020). Another focus is learning to align latent image-text embeddings, which allows to classify images using arbitrary texts. Frome et al. (2013) and Norouzi et al. (2014) are pioneering works that learn a visual-semantic embedding space using deep learning. Wang et al. (2018) distills information

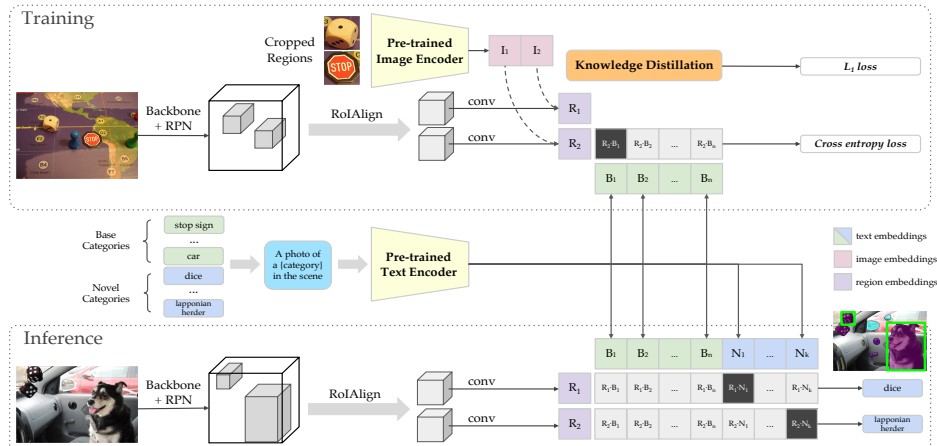

Figure 2: **An overview of using ViLD for open-vocabulary object detection.** ViLD distills the knowledge from a pretrained open-vocabulary image classification model. First, the category text embeddings and the image embeddings of cropped object proposals are computed, using the text and image encoders in the pretrained classification model. Then, ViLD employs the text embeddings as the region classifier (ViLD-text) and minimizes the distance between the region embedding and the image embedding for each proposal (ViLD-image). During inference, text embeddings of novel categories are used to enable open-vocabulary detection.

from both word embeddings and knowledge graphs. Recent work CLIP (Radford et al., 2021) and ALIGN (Jia et al., 2021) push the limit by collecting million-scale image-text pairs and then training joint image-text models using contrastive learning. These models can directly transfer to a suite of classification datasets and achieve impressive performances. While these work focus on image-level open-vocabulary recognition, we focus on detecting objects using arbitrary text inputs.

**Increasing vocabulary in object detection:** It's expensive to scale up the data collection for large vocabulary object detection. Zhao et al. (2020) and Zhou et al. (2021) unify the label space from multiple datasets. Joseph et al. (2021) incrementally learn identified unknown categories. Zero-shot detection (ZSD) offers another direction. Most ZSD methods align region features to pretrained text embeddings in base categories (Bansal et al., 2018; Demirel et al., 2018; Rahman et al., 2019; Hayat et al., 2020; Zheng et al., 2020). However, there is a large performance gap to supervised counterparts. To address this issue, Zareian et al. (2021) pretrain the backbone model using image captions and finetune the pretrained model with detection datasets. In contrast, we use an image-text pretrained model as a teacher model to supervise student object detectors. All previous methods are only evaluated on tens of categories, while we are the first to evaluate on more than 1,000 categories.

## 3 METHOD

**Notations:** We divide categories in a detection dataset into the base and novel subsets, and denote them by $C_B$ and $C_N$. Only annotations in $C_B$ are used for training. We use $\mathcal{T}(\cdot)$ to denote the text encoder and $\mathcal{V}(\cdot)$ to denote the image encoder in the pretrained open-vocabulary image classifier.

### 3.1 LOCALIZATION FOR NOVEL CATEGORIES

The first challenge for open-vocabulary detection is to localize novel objects. We modify a standard two-stage object detector, *e.g.*, Mask R-CNN (He et al., 2017), for this purpose. We replace its class-specific localization modules, *i.e.*, the second-stage bounding box regression and mask prediction layers, with class-agnostic modules for general object proposals. For each region of interest, these modules only predict a single bounding box and a single mask for all categories, instead of one prediction per category. The class-agnostic modules can generalize to novel objects.

### 3.2 OPEN-VOCABULARY DETECTION WITH CROPPED REGIONS

Once object candidates are localized, we propose to reuse a pretrained open-vocabulary image classifier to classify each region for detection.

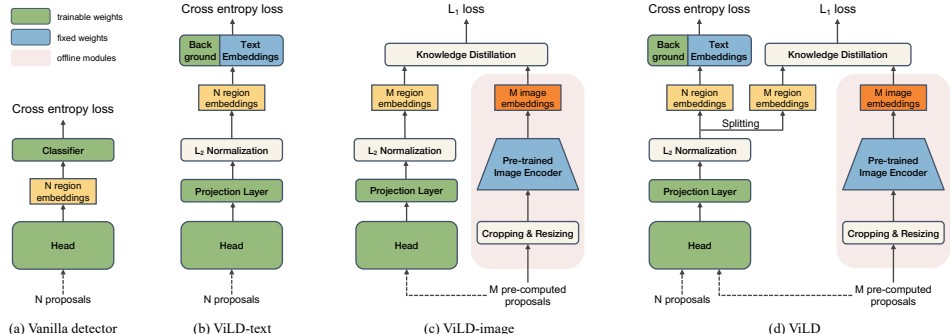

Figure 3: **Model architecture and training objectives.** **(a)** The classification head of a vanilla two-stage detector, *e.g.*, Mask R-CNN. **(b)** ViLD-text replaces the classifier with fixed text embeddings and a learnable background embedding. The projection layer is introduced to adjust the dimension of region embeddings to be compatible with the text embeddings. **(c)** ViLD-image distills from the precomputed image embeddings of proposals with an $\mathcal{L}_1$ loss. **(d)** ViLD combines ViLD-text and ViLD-image.

**Image embeddings:** We train a proposal network on base categories $C_B$ and extract the region proposals $\tilde{r} \in \widetilde{P}$ offline. We crop and resize the proposals, and feed them into the pretrained image encoder $\mathcal{V}$ to compute image embeddings $\mathcal{V}(\text{crop}(I, \tilde{r}))$, where $I$ is the image.

We ensemble the image embeddings from $1\times$ and $1.5\times$ crops, as the $1.5\times$ crop provides more context cues. The ensembled embedding is then renormalized to unit norm:

$$\mathcal{V}(\text{crop}(I, \tilde{r}_{\{1\times, 1.5\times\}})) = \frac{\mathbf{v}}{\|\mathbf{v}\|}, \text{ where } \mathbf{v} = \mathcal{V}(\text{crop}(I, \tilde{r}_{1\times})) + \mathcal{V}(\text{crop}(I, \tilde{r}_{1.5\times})). \quad (1)$$

**Text embeddings:** We generate the text embeddings offline by feeding the category texts with prompt templates, *e.g.*, "a photo of {category} in the scene", into the text encoder $\mathcal{T}$. We ensemble multiple prompt templates and the synonyms if provided.

Then, we compute cosine similarities between the image and text embeddings. A softmax activation is applied, followed by a per-class NMS to obtain final detections. The inference is slow since every cropped region is fed into $\mathcal{V}$.

## 3.3 ViLD: Vision and Language knowledge Distillation.

We propose **ViLD** to address the slow inference speed of the above method. **ViLD** learns region embeddings in a two-stage detector to represent each proposal $r$. We denote region embeddings by $\mathcal{R}(\phi(I), r)$, where $\phi(\cdot)$ is a backbone model and $\mathcal{R}(\cdot)$ is a lightweight head that generates region embeddings. Specifically, we take outputs before the classification layer as region embeddings.

**Replacing classifier with text embeddings:** We first introduce **ViLD-text**. Our goal is to train the region embeddings such that they can be classified by text embeddings. Fig. 3(b) shows the architecture and training objective. ViLD-text replaces the learnable classifier in Fig. 3(a) with the text embeddings introduced in Sec. 3.2. Only $\mathcal{T}(C_B)$, the text embeddings of $C_B$, are used for training. For the proposals that do not match any groundtruth in $C_B$, they are assigned to the background category. Since the text "background" does not well represent these unmatched proposals, we allow the background category to learn its own embedding $\mathbf{e}_{bg}$. We compute the cosine similarity between each region embedding $\mathcal{R}(\phi(I), r)$ and all category embeddings, including $\mathcal{T}(C_B)$ and $\mathbf{e}_{bg}$. Then we apply softmax activation with a temperature $\tau$ to compute the cross entropy loss. To train the first-stage region proposal network of the two-stage detector, we extract region proposals $r \in P$ online, and train the detector with ViLD-text from scratch. The loss for ViLD-text can be written as:

$$\mathbf{e}_r = \mathcal{R}(\phi(I), r)$$
$$\mathbf{z}(r) = \left[ sim(\mathbf{e}_r, \mathbf{e}_{bg}), \ sim(\mathbf{e}_r, \mathbf{t}_1), \ \cdots, \ sim(\mathbf{e}_r, \mathbf{t}_{|C_B|}) \right]$$
$$\mathcal{L}_{\text{ViLD-text}} = \frac{1}{N} \sum_{r \in P} \mathcal{L}_{\text{CE}}\left( softmax\big(\mathbf{z}(r)/\tau\big), y_r \right), \quad (2)$$

where $sim(\mathbf{a}, \mathbf{b}) = \mathbf{a}^\top\mathbf{b}/(\|\mathbf{a}\|\|\mathbf{b}\|)$, $\mathbf{t}_i$ denotes elements in $\mathcal{T}(C_B)$, $y_r$ denotes the class label of region $r$, $N$ is the number of proposals per image ($|P|$), and $\mathcal{L}_{CE}$ is the cross entropy loss.

During inference, we include novel categories ($C_N$) and generate $\mathcal{T}(C_B \cup C_N)$ (sometimes $\mathcal{T}(C_N)$ only) for open-vocabulary detection (Fig. 2). Our hope is that the model learned from annotations in $C_B$ can generalize to novel categories $C_N$.

**Distilling image embeddings:** We then introduce **ViLD-image**, which aims to distill the knowledge from the teacher image encoder $\mathcal{V}$ into the student detector. Specifically, we align region embeddings $\mathcal{R}(\phi(I), \tilde{r})$ to image embeddings $\mathcal{V}(\text{crop}(I, \tilde{r}))$ introduced in Sec. 3.2.

To make the training more efficient, we extract $M$ proposals $\tilde{r} \in \tilde{P}$ offline for each training image, and precompute the $M$ image embeddings. These proposals can contain objects in both $C_B$ and $C_N$, as the network can generalize. In contrast, ViLD-text can only learn from $C_B$. We apply an $\mathcal{L}_1$ loss between the region and image embeddings to minimize their distance. The ensembled image embeddings in Sec. 3.2 are used for distillation:

$$\mathcal{L}_{\text{ViLD-image}} = \frac{1}{M} \sum_{\tilde{r} \in \widetilde{P}} \|\mathcal{V}(\text{crop}(I, \tilde{r}_{\{1\times, 1.5\times\}})) - \mathcal{R}(\phi(I), \tilde{r})\|_1. \tag{3}$$

Fig. 3(c) shows the architecture. Zhu et al. (2019) use a similar approach to make Faster R-CNN features mimic R-CNN features, however, the details and goals are different: They reduce redundant context to improve supervised detection; while ViLD-image is to enable open-vocabulary detection on novel categories.

The total training loss of **ViLD** is simply a weighted sum of both objectives:

$$\mathcal{L}_{\text{ViLD}} = \mathcal{L}_{\text{ViLD-text}} + w \cdot \mathcal{L}_{\text{ViLD-image}}, \tag{4}$$

where $w$ is a hyperparameter weight for distilling the image embeddings. Fig. 3(d) shows the model architecture and training objectives. ViLD-image distillation only happens in training time. During inference, ViLD-image, ViLD-text and ViLD employ the same set of text embeddings as the detection classifier, and use the same architecture for open-vocabulary detection (Fig. 2).

### 3.4 MODEL ENSEMBLING

In this section, we explore model ensembling for the best detection performance over base and novel categories. First, we combine the predictions of a ViLD-text detector with the open-vocabulary image classification model. The intuition is that ViLD-image learns to approximate the predictions of its teacher model, and therefore, we assume using the teacher model directly may improve performance. We use a trained ViLD-text detector to obtain top $k$ candidate regions and their confidence scores. Let $p_{i,\text{ViLD-text}}$ denote the confidence score of proposal $\tilde{r}$ belonging to category $i$. We then feed $\text{crop}(I, \tilde{r})$ to the open-vocabulary classification model to obtain the teacher's confidence score $p_{i,\text{cls}}$. Since we know the two models have different performance on base and novel categories, we introduce a weighted geometric average for the ensemble:

$$p_{i,\text{ensemble}} = \begin{cases} p_{i,\text{ViLD-text}}^{\lambda} \cdot p_{i,\text{cls}}^{(1-\lambda)}, & \text{if } i \in C_B \\ p_{i,\text{ViLD-text}}^{(1-\lambda)} \cdot p_{i,\text{cls}}^{\lambda}. & \text{if } i \in C_N \end{cases} \tag{5}$$

$\lambda$ is set to $2/3$, which weighs the prediction of ViLD-text more on base categories and vice versa. Note this approach has a similar slow inference speed as the method in Sec. 3.2.

Next, we introduce a different ensembling approach to mitigate the above inference speed issue. Besides, in ViLD, the cross entropy loss of ViLD-text and the $\mathcal{L}_1$ distillation loss of ViLD-image is applied to the same set of region embeddings, which may cause contentions. Here, instead, we learn two sets of embeddings for ViLD-text (Eq. 2) and ViLD-image (Eq. 3) respectively, with two separate heads of identical architectures. Text embeddings are applied to these two regions embeddings to obtain confidence scores $p_{i,\text{ViLD-text}}$ and $p_{i,\text{ViLD-image}}$, which are then ensembled in the same way as Eq. 5, with $p_{i,\text{ViLD-image}}$ replacing $p_{i,\text{cls}}$. We name this approach **ViLD-ensemble**.

## 4 EXPERIMENTS

**Implementation details:** We benchmark on the Mask R-CNN (He et al., 2017) with ResNet (He et al., 2016) FPN (Lin et al., 2017) backbone and use the same settings for all models unless explicitly specified. The models use $1024 \times 1024$ as input image size, large-scale jittering augmentation of range $[0.1, 2.0]$, synchronized batch normalization (Ioffe & Szegedy, 2015; Girshick et al., 2018) of batch size 256, weight decay of 4e-5, and an initial learning rate of 0.32. We train the model from scratch for 180,000 iterations, and divide the learning rate by 10 at $0.9\times$, $0.95\times$, and $0.975\times$ of total iterations. We use the publicly available pretrained CLIP model[1] as the open-vocabulary classification model, with an input size of $224 \times 224$. The temperature $\tau$ is set to 0.01, and the maximum number of detections per image is 300. We refer the readers to Appendix D for more details.

### 4.1 BENCHMARK SETTINGS

We mainly evaluate on LVIS (Gupta et al., 2019) with our new setting. To compare with previous methods, we also use the setting in Zareian et al. (2021), which is adopted in many zero-shot detection works.

**LVIS:** We benchmark on LVIS v1. LVIS contains a large and diverse set of vocabulary (1,203 categories) that is more suitable for open-vocabulary detection. We take its 866 frequent and common categories as the base categories $C_B$, and hold out the 337 rare categories as the novel categories $C_N$. $AP_r$, the AP of rare categories, is the main metric.

**COCO:** Bansal et al. (2018) divide COCO-2017 (Lin et al., 2014) into 48 base categories and 17 novel categories, removing 15 categories without a synset in the WordNet hierarchy. We follow previous works and do not compute instance masks. We evaluate on the generalized setting.

### 4.2 LEARNING GENERALIZABLE OBJECT PROPOSALS

We first study whether a detector can localize novel categories when only trained on base categories. We evaluate the region proposal networks in Mask R-CNN with a ResNet-50 backbone. Table 1 shows the average recall (AR) (Lin et al., 2014) on novel categories. Training with only base categories performs slightly worse by $\sim 2$ AR at 100, 300, and 1000 proposals, compared to using both base and novel categories. This experiment demonstrates that, without seeing novel categories during training, region proposal networks can generalize to novel categories, only suffering a small performance drop. We believe better proposal networks focusing on unseen category generalization should further improve the performance, and leave this for future research.

Table 1: **Training with only base categories achieves comparable average recall (AR) for novel categories on LVIS.** We compare RPN trained with base only *vs.* base+novel categories and report the bounding box AR.

| Supervision | $AR_r$@100 | $AR_r$@300 | $AR_r$@1000 |
|---|---|---|---|
| base | 39.3 | 48.3 | 55.6 |
| base + novel | 41.1 | 50.9 | 57.0 |

### 4.3 OPEN-VOCABULARY CLASSIFIER ON CROPPED REGIONS

In Table 2, we evaluate the approach in Sec. 3.2, *i.e.*, using an open-vocabulary classifier to classify cropped region proposals. We use CLIP in this experiment and find it tends to output confidence scores regardless of the localization quality (Appendix B). Given that, we ensemble the CLIP confidence score with a proposal objectness score by geometric mean. Results show it improves both base and novel APs. We compare with supervised baselines trained on base/base+novel categories, as well as Supervised-RFS (Mahajan et al., 2018; Gupta et al., 2019) that uses category frequency for balanced sampling. CLIP on cropped regions already outperforms supervised baselines on $AP_r$ by a large margin, without accessing detection annotations in novel categories. However, the performances of $AP_c$ and $AP_f$ are still trailing behind. This experiment shows that a strong open-vocabulary classification model can be a powerful teacher model for detecting novel objects, yet there is still much improvement space for inference speed and overall AP.

---

[1] https://github.com/openai/CLIP, ViT-B/32.

Table 2: **Using CLIP for open-vocabulary detection achieves high detection performance on novel categories.** We apply CLIP to classify cropped region proposals, with or without ensembling objectness scores, and report the mask average precision (AP). The performance on novel categories ($AP_r$) is far beyond supervised learning approaches. However, the overall performance is still behind.

| Method | $AP_r$ | $AP_c$ | $AP_f$ | AP |
|---|---|---|---|---|
| Supervised (base class only) | 0.0 | 22.6 | 32.4 | 22.5 |
| CLIP on cropped regions w/o objectness | 13.0 | 10.6 | 6.0 | 9.2 |
| CLIP on cropped regions | **18.9** | 18.8 | 16.0 | 17.7 |
| Supervised (base+novel) | 4.1 | 23.5 | 33.2 | 23.9 |
| Supervised-RFS (base+novel) | 12.3 | 24.3 | 32.4 | 25.4 |

Table 3: **Performance of ViLD and its variants. ViLD outperforms the supervised counterpart on novel categories.** Using ALIGN as the teacher model achieves the best performance without bells and whistles. All results are mask AP. We average over 3 runs for R50 experiments. †: methods with R-CNN style; runtime is 630× of Mask R-CNN style. ‡: for reference, fully-supervised learning with additional tricks.

| Backbone | Method | $AP_r$ | $AP_c$ | $AP_f$ | AP |
|---|---|---|---|---|---|
| ResNet-50+ViT-B/32 | CLIP on cropped regions† | 18.9 | 18.8 | 16.0 | 17.7 |
| | ViLD-text+CLIP† | **22.6** | 24.8 | 29.2 | 26.1 |
| ResNet-50 | Supervised-RFS (base+novel) | 12.3 | 24.3 | 32.4 | 25.4 |
| | GloVe baseline | 3.0 | 20.1 | 30.4 | 21.2 |
| | ViLD-text | 10.1 | 23.9 | 32.5 | 24.9 |
| | ViLD-image | 11.2 | 11.3 | 11.1 | 11.2 |
| | ViLD ($w$=0.5) | 16.1 | 20.0 | 28.3 | 22.5 |
| | ViLD-ensemble ($w$=0.5) | **16.6** | 24.6 | 30.3 | 25.5 |
| EfficientNet-b7 | ViLD-ensemble w/ ViT-L/14 ($w$=1.0) | 21.7 | 29.1 | 33.6 | 29.6 |
| | ViLD-ensemble w/ ALIGN ($w$=1.0) | **26.3** | 27.2 | 32.9 | 29.3 |
| ResNeSt269+HTC | 2020 Challenge winner (Tan et al., 2020)‡ | 30.0 | 41.9 | 46.0 | 41.5 |

## 4.4 VISION AND LANGUAGE KNOWLEDGE DISTILLATION

We evaluate the performance of ViLD and its variants (ViLD-text, ViLD-image, and ViLD-ensemble), which are significantly faster compared to the method in Sec. 4.3. Finally, we use stronger teacher models to demonstrate our best performance. Table 3 summarizes the results.

**Text embeddings as classifiers (ViLD-text):** We evaluate ViLD-text using text embeddings generated by CLIP, and compare it with GloVe text embeddings (Pennington et al., 2014) pretrained on a large-scale text-only corpus. Table 3 shows ViLD-text achieves 10.1 $AP_r$, which is significantly better than 3.0 $AP_r$ using GloVe. This demonstrates the importance of using text embeddings that are jointly trained with images. ViLD-text achieves much higher $AP_c$ and $AP_f$ compared to CLIP on cropped regions (Sec. 4.3), because ViLD-text uses annotations in $C_B$ to align region embeddings with text embeddings. The $AP_r$ is worse, showing that using only 866 base categories in LVIS does not generalize as well as CLIP to novel categories.

**Distilling image embeddings (ViLD-image):** We evaluate ViLD-image, which distills from the image embeddings of cropped region proposals, inferred by CLIP's image encoder, with a distillation weight of 1.0. Experiments show that ensembling with objectness scores doesn't help with other ViLD variants, so we only apply it to ViLD-image. Without training with any object category labels, ViLD-image achieves 11.2 $AP_r$ and 11.2 overall AP. This demonstrates that visual distillation works for open-vocabulary detection but the performance is not as good as CLIP on cropped regions.

**Text+visual embeddings (ViLD):** ViLD shows the benefits of combining distillation loss (ViLD-image) with classification loss using text embeddings (ViLD-text). We explore different hyperparameter settings in Appendix Table 7 and observe a consistent trade-off between $AP_r$ and $AP_{c,f}$, which suggests there is a competition between ViLD-text and ViLD-image. In Table 3, we compare ViLD with other methods. Its $AP_r$ is 6.0 higher than ViLD-text and 4.9 higher than ViLD-image, indicating combining the two learning objectives boosts the performance on novel categories. ViLD outperforms Supervised-RFS by 3.8 $AP_r$, showing our open-vocabulary detection approach is better than supervised models on rare categories.

Table 4: **Performance on COCO dataset compared with existing methods.** ViLD outperforms all the other methods in the table trained with various sources by a large margin, on both novel and base categories.

| Method | Training source | Novel AP | Base AP | Overall AP |
|---|---|---|---|---|
| Bilen & Vedaldi (2016) | image-level labels in $C_B \cup C_N$ | 19.7 | 19.6 | 19.6 |
| Ye et al. (2019) | | 20.3 | 20.1 | 20.1 |
| Bansal et al. (2018) | | 0.31 | 29.2 | 24.9 |
| Zhu et al. (2020) | instance-level labels in $C_B$ | 3.41 | 13.8 | 13.0 |
| Rahman et al. (2020) | | 4.12 | 35.9 | 27.9 |
| Zareian et al. (2021) | image captions in $C_B \cup C_N$ instance-level labels in $C_B$ | 22.8 | 46.0 | 39.9 |
| CLIP on cropped regions | | 26.3 | 28.3 | 27.8 |
| ViLD-text | image-text pairs from Internet | 5.9 | 61.8 | 47.2 |
| ViLD-image | (may contain $C_B \cup C_N$) | 24.1 | 34.2 | 31.6 |
| ViLD ($w = 0.5$) | instance-level labels in $C_B$ | **27.6** | 59.5 | 51.3 |

**Model ensembling:** We study methods discussed in Sec. 3.4 to reconcile the conflict of joint training with ViLD-text and ViLD-image. We use two ensembling approaches: 1) ensembling ViLD-text with CLIP (**ViLD-text+CLIP**); 2) ensembling ViLD-text and ViLD-image using separate heads (**ViLD-ensemble**). As shown in Table 3, ViLD-ensemble improves performance over ViLD, mainly on $AP_c$ and $AP_r$. This shows ensembling reduces the competition. ViLD-text+CLIP obtains much higher $AP_r$, outperforming ViLD by 6.5, and maintains good $AP_{c,f}$. Note that it is slow and impractical for real world applications. This experiment is designed for showing the potential of using open-vocabulary classification models for open-vocabulary detection.

**Stronger teacher model:** We use CLIP ViT-L/14 and ALIGN (Jia et al., 2021) to explore the performance gain with a stronger teacher model (details in Appendix D). As shown in Table 3, both models achieve superior results compared with R50 ViLD w/ CLIP. The detector distilled from ALIGN is only trailing to the fully-supervised 2020 Challenge winner (Tan et al., 2020) by 3.7 $AP_r$, which employs two-stage training, self-training, and multi-scale testing *etc*. The results demonstrate ViLD scales well with the teacher model, and is a promising open-vocabulary detection approach.

## 4.5 PERFORMANCE COMPARISON ON COCO DATASET

Several related works in zero-shot detection and open-vocabulary detection are evaluated on COCO. To compare with them, we train and evaluate ViLD variants following the benchmark setup in Zareian et al. (2021) and report box AP with an IoU threshold of 0.5. We use the ResNet-50 backbone, shorten the training schedule to 45,000 iterations, and keep other settings the same as our experiments on LVIS. Table 4 summarizes the results. ViLD outperforms Zareian et al. (2021) by 4.8 Novel AP and 13.5 Base AP. Different from Zareian et al. (2021), we do not have a pretraining phase tailored for detection. Instead, we use an off-the-shelf classification model. The performance of ViLD-text is low because only 48 base categories are available, which makes generalization to novel categories challenging. In contrast, ViLD-image and ViLD, which can distill image features of novel categories, outperform all existing methods (not apple-to-apple comparison though, given different methods use different settings).

## 4.6 TRANSFER TO OTHER DATASETS

Trained ViLD models can be transferred to other detection datasets, by simply switching the classifier to the category text embeddings of the new datasets. For simplicity, we keep the background embedding trained on LVIS. We evaluate the transferability of ViLD on PASCAL VOC (Everingham et al., 2010), COCO (Lin et al., 2014), and Objects365 (Shao et al., 2019). Since the three datasets have much smaller vocabularies, category overlap is unavoidable and images can be shared among datasets, *e.g.*, COCO and LVIS. As shown in Table 5, ViLD achieves better transfer performance than ViLD-text. In PASCAL and COCO, the gap is large. This improvement should be credited to visual distillation, which better aligns region embeddings with the text classifier. We also compare with supervised learning and finetuning the classification layer. Although across datasets, ViLD has 3-6 AP gaps compared to the finetuning method and larger gaps compared to the supervised method, it is the first time we can directly transfer a trained detector to different datasets using language.

Table 5: **Generalization ability of ViLD.** We evaluate the LVIS-trained model with ResNet-50 backbone on PASCAL VOC 2007 test set, COCO validation set, and Objects365 v1 validation set. Simply replacing the text embeddings, our approaches are able to transfer to various detection datasets. The supervised baselines of COCO and Objects365 are trained from scratch. [†]: the supervised baseline of PASCAL VOC is initialized with an ImageNet-pretrained checkpoint. All results are box APs.

| Method | PASCAL VOC[†] | | COCO | | | Objects365 | | |
|---|---|---|---|---|---|---|---|---|
| | $AP_{50}$ | $AP_{75}$ | AP | $AP_{50}$ | $AP_{75}$ | AP | $AP_{50}$ | $AP_{75}$ |
| ViLD-text | 40.5 | 31.6 | 28.8 | 43.4 | 31.4 | 10.4 | 15.8 | 11.1 |
| ViLD | 72.2 | 56.7 | 36.6 | 55.6 | 39.8 | 11.8 | 18.2 | 12.6 |
| Finetuning | 78.9 | 60.3 | 39.1 | 59.8 | 42.4 | 15.2 | 23.9 | 16.2 |
| Supervised | 78.5 | 49.0 | 46.5 | 67.6 | 50.9 | 25.6 | 38.6 | 28.0 |

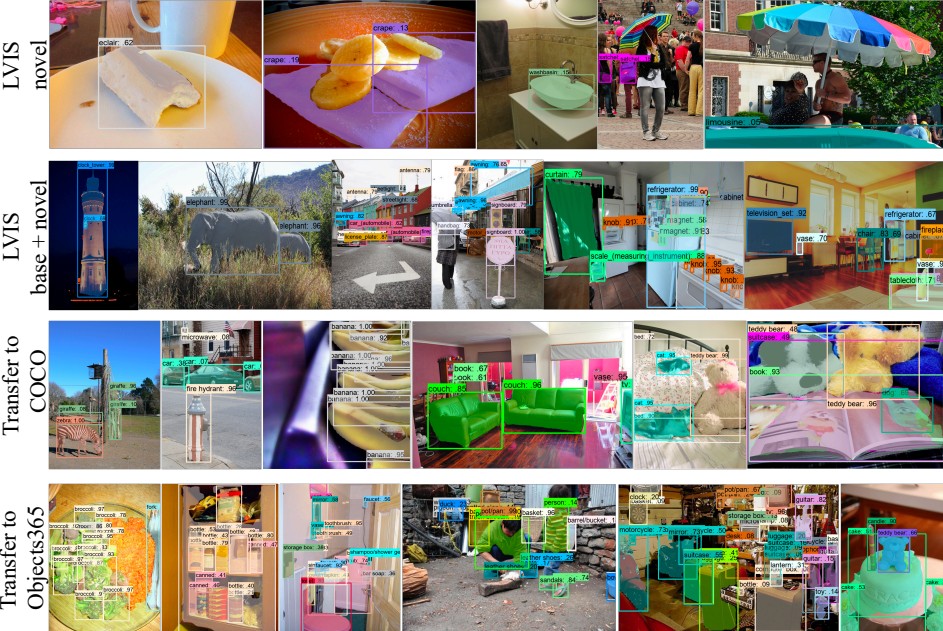

Figure 4: **Qualitative results on LVIS, COCO, and Objects365.** First row: ViLD is able to correctly localize and recognize objects in novel categories. For clarity, we only show the detected novel objects. Second row: The detected objects on base+novel categories. The performance on base categories is not degraded with ViLD. Last two rows: ViLD can directly transfer to COCO and Objects365 without further finetuning.

## 4.7 QUALITATIVE RESULTS

In Fig. 4, we visualize ViLD's detection results. It illustrates ViLD is able to detect objects of both novel and base categories, with high-quality mask predictions on novel objects, *e.g.*, it well separates banana slices from the crepes (novel category). We also show qualitative results on COCO and Objects365, and find ViLD generalizes well. We show more qualitative results, *e.g.*, interactive detection and systematic expansion, in Appendix A.

## 5 CONCLUSION

We present ViLD, an open-vocabulary object detection method by distilling knowledge from open-vocabulary image classification models. ViLD is the first open-vocabulary detection method evaluated on the challenging LVIS dataset. It attains 16.1 AP for novel cateogires on LVIS with a ResNet50 backbone, which surpasses its supervised counterpart at the same inference speed. With a stronger teacher model (ALIGN), the performance can be further improved to 26.3 novel AP. We demonstrate that the detector learned from LVIS can be directly transferred to 3 other detection datasets. We hope that the simple design and strong performance make ViLD a scalable alternative approach for detecting long-tailed categories, instead of collecting expensive detection annotations.

ETHICS STATEMENT

Our paper studies open-vocabulary object detection, a sub-field in computer vision. Our method is based on knowledge distillation, a machine learning technique that has been extensively used in computer vision, natural language processing, *etc*. All of our experiments were conducted on public datasets with pretrained models that are either publicly available or introduced in published papers. The method proposed in our paper is a principled method for open-vocabulary object detection that can be used in a wide range of applications. Therefore, the ethical impact of our work would primarily depends on the specific applications. We foresee positive impacts if our method is applied to object detection problems where the data collection is difficult to scale, such as detecting rare objects for self-driving cars. But the method can also be applied to other sensitive applications that could raise ethical concerns, such as video surveillance systems.

REPRODUCIBILITY STATEMENT

We provide detailed descriptions of the proposed method in Sec. 3. Details about experiment settings, hyper-parameters and implementations are presented in Sec. 4, Appendix C and Appendix D. We release our code and pretrained models at `https://github.com/tensorflow/tpu/tree/master/models/official/detection/projects/vild` to facilitate the reproducibility of our work.

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

APPENDIX

## A    ADDITIONAL QUALITATIVE RESULTS

**On-the-fly interactive object detection:**  We tap the potential of ViLD by using arbitrary text to interactively recognize fine-grained categories and attributes. We extract the region embedding and compute its cosine similarity with a small set of on-the-fly arbitrary texts describing attributes and/or fine-grained categories; we apply softmax with temperature $\tau$ on top of the similarities. To our surprise, though never trained on fine-grained dog breeds (Fig. 5), it correctly distinguishes husky from shiba inu. It also works well on identifying object colors (Fig. 1). The results demonstrate knowledge distillation from an open-vocabulary image classification model helps ViLD to gain understanding of concepts not present in the detection training. Of course, ViLD does not work all the time, *e.g.*, it fails to recognize poses of animals.

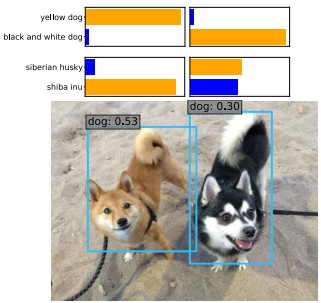

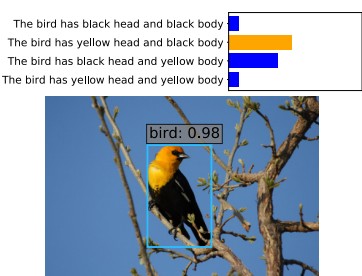

(a) Fine-grained breeds and colors.        (b) Colors of body parts.

Figure 5: **On-the-fly interactive object detection.** One application of ViLD is using on-the-fly arbitrary texts to further recognize more details of the detected objects, *e.g.*, fine-grained categories and color attributes.

**Systematic expansion of dataset vocabulary:**  In addition, we propose to systematically expand the dataset vocabulary ($\mathbf{v} = \{v_1, ..., v_p\}$) with a set of attributes ($\mathbf{a} = \{a_1, ..., a_q\}$) as follows:

$$\Pr(v_i, a_j \mid \mathbf{e}_r) = \Pr(v_i \mid \mathbf{e}_r) \cdot \Pr(a_j \mid v_i, \mathbf{e}_r)$$
$$= \Pr(v_i \mid \mathbf{e}_r) \cdot \Pr(a_j \mid \mathbf{e}_r), \qquad (6)$$

where $\mathbf{e}_r$ denotes the region embedding. We assume $v_i \perp\!\!\!\perp a_j \mid \mathbf{e}_r$, *i.e.*, given $\mathbf{e}_r$ the event the object belongs to category $v_i$ is conditionally independent to the event it has attribute $a_j$.

Let $\tau$ denote the temperature used for softmax and $\mathcal{T}$ denote the text encoder as in Eq. 2. Then

$$\Pr(v_i \mid \mathbf{e}_r) = softmax_i(sim(\mathbf{e}_r, \mathcal{T}(\mathbf{v}))/\tau), \qquad (7)$$
$$\Pr(a_j \mid \mathbf{e}_r) = softmax_j(sim(\mathbf{e}_r, \mathcal{T}(\mathbf{a}))/\tau). \qquad (8)$$

In this way, we are able to expand $p$ vocabularies into a new set of $p \times q$ vocabularies with attributes. The conditional probability approach is similar to YOLO9000 (Redmon & Farhadi, 2017). We show a qualitative example of this approach in Fig. 6, where we use a color attribute set as $\mathbf{a}$. Our open-vocabulary detector successfully detects fruits with color attributes.

We further expand the detection vocabulary to fine-grained bird categories by using all 200 species from CUB-200-2011 (Wah et al., 2011). Fig. 7 shows successful and failure examples of our open-vocabulary fine-grained detection on CUB-200-2011 images. In general, our model is able to detect visually distinctive species, but fails at other ones.

**Transfer to PASCAL VOC:**  In Fig. 8, we show qualitative results of transferring an open-vocabulary detector trained on LVIS (Gupta et al., 2019) to PASCAL VOC Detection (2007 test set) (Everingham et al., 2010), without finetuning (Sec. 4.6 in the main paper). Results demonstrate that the transferring works well.

**Failure cases:**  In Fig. 9, we show two failure cases of ViLD. The most common failure cases are the missed detection. A less common mistake is misclassifying the object category.

We show a failure case of mask prediction on PASCAL VOC in Fig. 10. It seems that the mask prediction is sometimes based on low-level appearance rather than semantics.

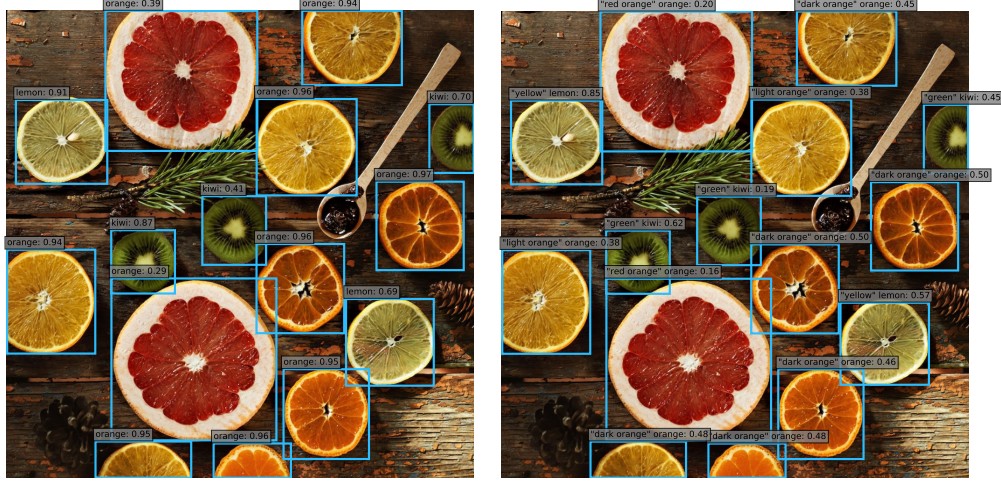

(a) Using LVIS vocabulary.  (b) After expanding vocabulary with color attributes.

Figure 6: **Systematic expansion of dataset vocabulary with colors.** We add 11 color attributes (*red orange, dark orange, light orange, yellow, green, cyan, blue, purple, black, brown, white*) to LVIS categories, which expand the vocabulary size by 11×. Above we show an example of detection results. Our open-vocabulary detector is able to assign the correct color to each fruit. A class-agnostic NMS with threshold 0.9 is applied. Each figure shows top 15 predictions.

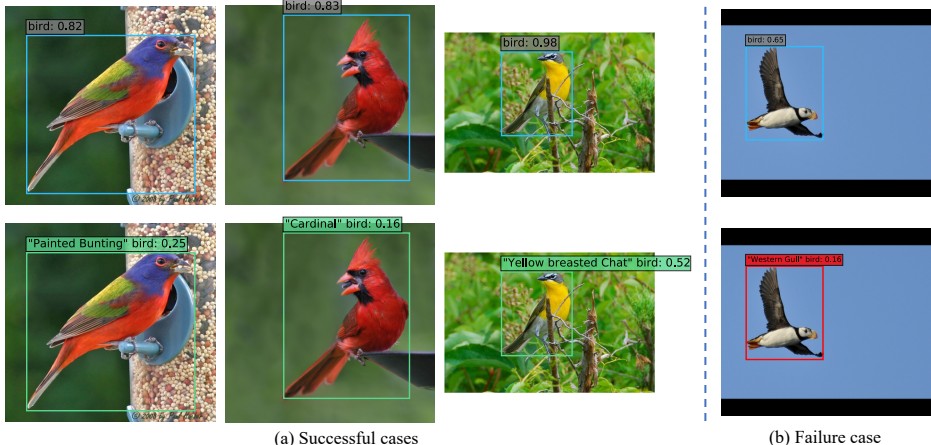

(a) Successful cases  (b) Failure case

Figure 7: **Systematic expansion of dataset vocabulary with fine-grained categories.** We use the systematic expansion method to detect 200 fine-grained bird species in CUB-200-2011. **(a)**: Our open-vocabulary detector is able to perform fine-grained detection **(bottom)** using the detector trained on LVIS **(top)**. **(b)**: It fails at recognizing visually non-distinctive species. It incorrectly assigns "Western Gull" to "Horned Puffin" due to visual similarity.

# B  ANALYSIS OF CLIP ON CROPPED REGIONS

In this section, we analyze some common failure cases of CLIP on cropped regions and discuss possible ways to mitigate these problems.

**Visual similarity:**  This confusion is common for any classifiers and detectors, especially on large vocabularies. In Fig. 11(a), we show two failure examples due to visual similarity. Since we only use a relatively small ViT-B/32 CLIP model, potentially we can improve the performance with a higher-capacity pretrained model. In Table 6, when replacing this CLIP model with an EfficientNet-l2 ALIGN model, we see an increase on AP.

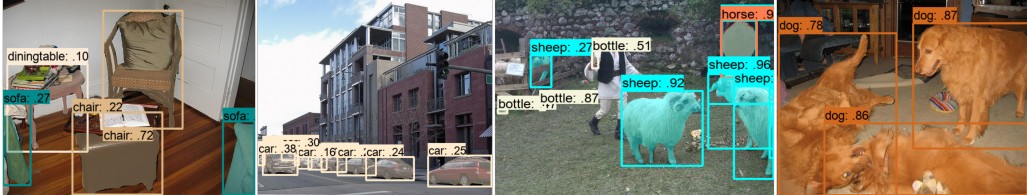

Figure 8: **Transfer to PASCAL VOC.** ViLD correctly detects objects when transferred to PASCAL VOC, where images usually have lower resolution than LVIS (our training set). In the third picture, our detector is able to find tiny bottles, though it fails to detect the person.

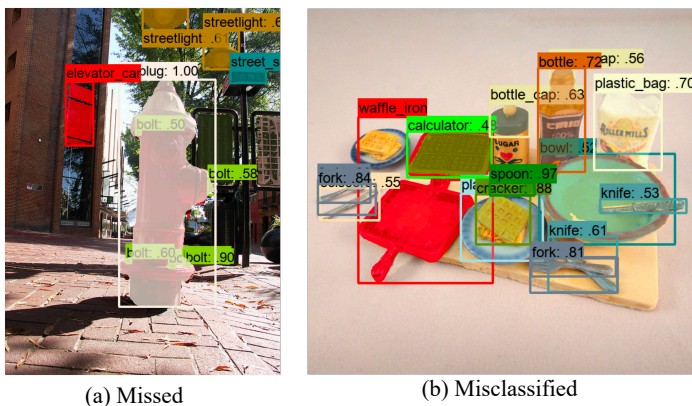

(a) Missed        (b) Misclassified

Figure 9: **Failure cases on LVIS novel categories.** The red bounding boxes indicate the groundtruths of the failed detections. **(a)** A common failure type where the novel objects are missing, *e.g.*, the elevator car is not detected. **(b)** A less common failure where (part of) the novel objects are misclassified, *e.g.*, half of the waffle iron is detected as a calculator due to visual similarity.

**Aspect ratio:** This issue is introduced by the pre-processing of inputs in CLIP. We use the ViT-B/32 CLIP with a fixed input resolution of $224 \times 224$. It resizes the shorter edge of the image to 224, and then uses a center crop. However, since region proposals can have more extreme aspect ratios than the training images for CLIP, and some proposals are tiny, we directly resize the proposals to that resolution, which might cause some issues. For example, the thin structure in Fig. 11(b) right will be highly distorted with the pre-processing. And the oven and fridge can be confusing with the distorted aspect ratio. There might be some simple remedies for this, *e.g.*, pasting the cropped region with original aspect ratio on a black background. We tried this simple approach with both CLIP and ALIGN. Preliminary results show that it works well on the fully convolutional ALIGN, while doesn't work well on the transformer-based CLIP, probably because CLIP is never trained with black image patches.

**Multiple objects in a bounding box:** Multiple objects in a region interfere CLIP's classification results, see Fig. 11(c), where a corner of an aquarium dominates the prediction. This is due to CLIP pretraining, which pairs an entire image with its caption. The caption is usually about salient objects in the image. It's hard to mitigate this issue at the open-vocabulary classification model's end. On the other hand, a supervised detector are trained to recognize the object tightly surrounded by the bounding box. So when distilling knowledge from an open-vocabulary image classification model, keeping training a supervised detector on base categories could help, as can be seen from the improvement of ViLD over ViLD-image (Sec. 4.4).

**Confidence scores predicted by CLIP do not reflect the localization quality:** For example, in Fig. 12(a), CLIP correctly classifies the object, but gives highest scores to partial detection boxes. CLIP is not trained to measure the quality of bounding boxes. Nonetheless, in object detection, it is important for the higher-quality boxes to have higher scores. In Fig. 12(c), we simply re-score by taking the geometric mean of the CLIP confidence score and the objectness score from the proposal model, which yields much better top predictions. In Fig. 12(b), we show top predictions of the Mask

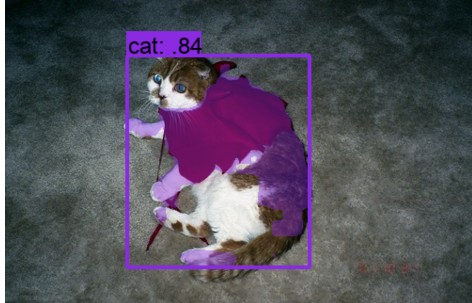

Figure 10: **An example of ViLD on PASCAL VOC showing a mask of poor quality.** The class-agnostic mask prediction head occasionally predicts masks based on low-level appearance rather than semantics, and thus fails to obtain a complete instance mask.

Table 6: **ALIGN on cropped regions achieves superior AP$_r$, and overall very good performance.** It shows a stronger open-vocabulary classification model can improve detection performance by a large margin. We report box APs here.

| Method | AP$_r$ | AP$_c$ | AP$_f$ | AP |
|---|---|---|---|---|
| CLIP on cropped regions | 19.5 | 19.7 | 17.0 | 18.6 |
| ALIGN on cropped regions | **39.6** | 32.6 | 26.3 | 31.4 |

R-CNN model. Its top predictions have good bounding boxes, while the predicted categories are wrong. This experiment shows that it's important to have both an open-vocabulary classification model for better recognition, as well as supervision from detection dataset for better localization.

## C  ADDITIONAL QUANTITATIVE RESULTS

**Hyperparameter sweep for visual distillation:**  Table 7 shows the parameter sweep of different distillation weights using $\mathcal{L}_1$ and $\mathcal{L}_2$ losses. Compared with no distillation, additionally learning from image embeddings generally yields better performance on novel categories. We find $\mathcal{L}_1$ loss can better improve the AP$_r$ performance with the trade-off against AP$_c$ and AP$_f$. This suggests there is a competition between ViLD-text and ViLD-image.

Table 7: **Hyperparameter sweep for visual distillation in ViLD.** $\mathcal{L}_1$ loss is better than $\mathcal{L}_2$ loss. For $\mathcal{L}_1$ loss, there is a trend that AP$_r$ increases as the weight increases, while AP$_{f,c}$ decrease. For all parameter combinations, ViLD outperforms ViLD-text on AP$_r$. We use ResNet-50 backbone and shorter training iterations (84,375 iters), and report mask AP in this table.

| Distill loss | Distill weight $w$ | AP$_r$ | AP$_c$ | AP$_f$ | AP |
|---|---|---|---|---|---|
| No distill | 0.0 | 10.4 | 22.9 | 31.3 | 24.0 |
| $\mathcal{L}_2$ loss | 0.5 | **13.7** | 21.7 | 31.2 | 24.0 |
| | 1.0 | 12.4 | 22.7 | 31.4 | 24.3 |
| | 2.0 | 13.4 | 22.0 | 30.9 | 24.0 |
| $\mathcal{L}_1$ loss | 0.05 | 12.9 | 22.4 | 31.7 | 24.4 |
| | 0.1 | 14.0 | 20.9 | 31.2 | 23.8 |
| | 0.5 | 16.3 | 19.2 | 27.3 | 21.9 |
| | 1.0 | **17.3** | 18.2 | 25.1 | 20.7 |

**Box APs and ResNet-152 backbone:**  Table 8 shows the corresponding box AP of Table 3 in the main paper. In general, box AP is slightly higher than mask AP. In addition, we include the results of ViLD variants with the ResNet-152 backbone. The deeper backbone improves all metrics. The trend/relative performance is consistent for box and mask APs, as well as for different backbones. ViLD-ensemble achieves the best box and mask AP$_r$.

**Ablation study on prompt engineering:**  We conduct an ablation study on prompt engineering. We compare the text embeddings ensembled over synonyms and 63 prompt templates (listed in Appendix D) with a non-ensembled version: Using the single prompt template "a photo of {article}

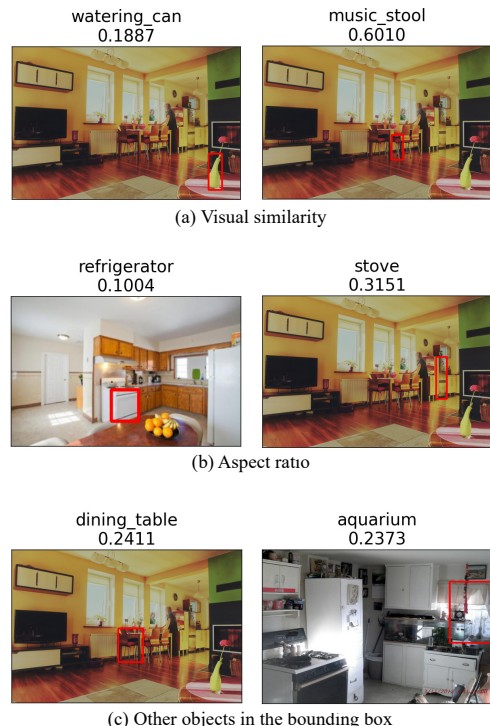

Figure 11: **Typical errors of CLIP on cropped regions.** **(a):** The prediction and the groundtruth have high visual similarity. **(b):** Directly resizing the cropped regions changes the aspect ratios, which may cause troubles. **(c):** CLIP's predictions are sometimes affected by other objects appearing in the region, rather than predicting what the entire bounding box is.

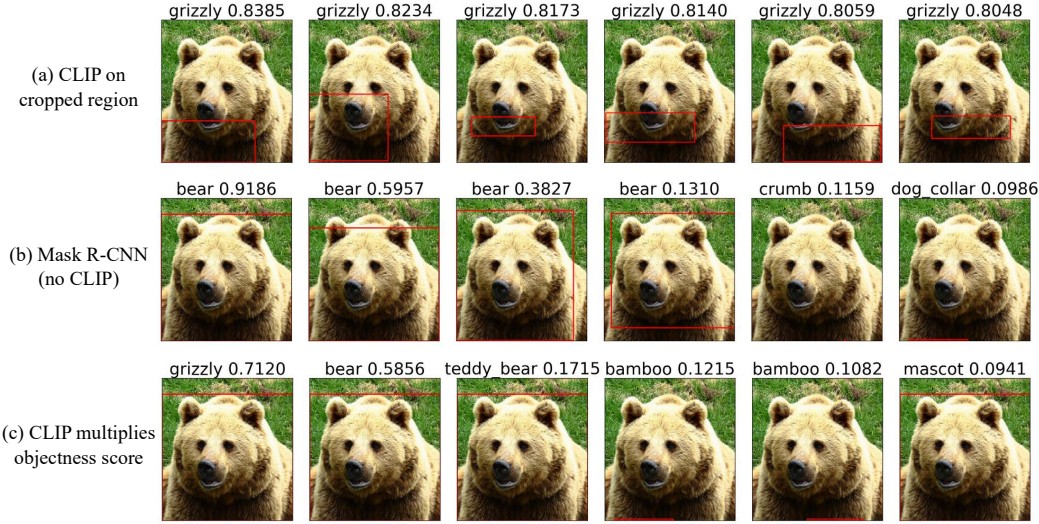

Figure 12: **The prediction scores of CLIP do not reflect the quality of bounding box localization. (a):** Top predictions of CLIP on cropped region. Boxes of poor qualities receive high scores, though the classification is correct. **(b):** Top predictions of a vanilla Mask R-CNN model. Box qualities are good while the classification is wrong. **(c):** We take the geometric mean of CLIP classification score and objectiveness score, and use it to rescore (a). In this way, a high-quality box as well as the correct category rank first.

{category}". Table 9 illustrates that ensembling multiple prompts slightly improves the performance by 0.4 $AP_r$.

Table 8: **Performance of ViLD variants.** This table shows additional box APs for models in Table 3 and ResNet-152 results.

| Backbone | Method | Box | | | | Mask | | | |
|---|---|---|---|---|---|---|---|---|---|
| | | $AP_r$ | $AP_c$ | $AP_f$ | AP | $AP_r$ | $AP_c$ | $AP_f$ | AP |
| ResNet-50 +ViT-B/32 | CLIP on cropped regions | 19.5 | 19.7 | 17.0 | 18.6 | 18.9 | 18.8 | 16.0 | 17.7 |
| | ViLD-text+CLIP | **23.8** | 26.7 | 32.8 | 28.6 | **22.6** | 24.8 | 29.2 | 26.1 |
| ResNet-50 | Supervised-RFS (base+novel) | 13.0 | 26.7 | 37.4 | 28.5 | 12.3 | 24.3 | 32.4 | 25.4 |
| | GloVe baseline | 3.2 | 22.0 | 34.9 | 23.8 | 3.0 | 20.1 | 30.4 | 21.2 |
| | ViLD-text | 10.6 | 26.1 | 37.4 | 27.9 | 10.1 | 23.9 | 32.5 | 24.9 |
| | ViLD-image | 10.3 | 11.5 | 11.1 | 11.2 | 11.2 | 11.3 | 11.1 | 11.2 |
| | ViLD ($w$=0.5) | 16.3 | 21.2 | 31.6 | 24.4 | 16.1 | 20.0 | 28.3 | 22.5 |
| | ViLD-ensemble ($w$=0.5) | **16.7** | 26.5 | 34.2 | 27.8 | **16.6** | 24.6 | 30.3 | 25.5 |
| ResNet-152 | Supervised-RFS (base+novel) | 16.2 | 29.6 | 39.7 | 31.2 | 14.4 | 26.8 | 34.2 | 27.6 |
| | ViLD-text | 12.3 | 28.3 | 39.7 | 30.0 | 11.7 | 25.8 | 34.4 | 26.7 |
| | ViLD-image | 12.5 | 13.9 | 13.4 | 13.4 | 13.1 | 13.4 | 13.0 | 13.2 |
| | ViLD ($w$=1.0) | 19.1 | 22.4 | 31.5 | 25.4 | **18.7** | 21.1 | 28.4 | 23.6 |
| | ViLD-ensemble ($w$=2.0) | **19.8** | 27.1 | 34.5 | 28.7 | **18.7** | 24.9 | 30.6 | 26.0 |
| EfficientNet-b7 | ViLD-ensemble w/ ViT-L/14 ($w$=1.0) | 22.0 | 31.5 | 38.0 | 32.4 | 21.7 | 29.1 | 33.6 | 29.6 |
| | ViLD-ensemble w/ ALIGN ($w$=1.0) | **27.0** | 29.4 | 36.5 | 31.8 | **26.3** | 27.2 | 32.9 | 29.3 |

Table 9: **Ablation study on prompt engineering.** Results indicate ensembling multiple prompt templates slightly improves $AP_r$. ViLD w/ multiple prompts is the same ViLD model in Table 3, and ViLD w/ single prompt only changes the text embeddings used as the classifier.

| Method | $AP_r$ | $AP_c$ | $AP_f$ | AP |
|---|---|---|---|---|
| ViLD w/ single prompt | 15.7 | 19.7 | 28.9 | 22.6 |
| ViLD w/ multiple prompts | **16.1** | 20.0 | 28.3 | 22.5 |

# D   MORE IMPLEMENTATION DETAILS

**ViLD-ensemble architecture:** In Fig. 13, we show the detailed architecture and learning objectives for ViLD-ensemble, the ensembling technique introduced in Sec. 3.4.

**Model used for qualitative results:** For all qualitative results, we use a ViLD model with ResNet-152 backbone, whose performance is shown in Table 8.

**Details for supervised baselines:** For a fair comparison, we train the second stage box/mask prediction heads of Supervised and Supervised-RFS baselines in the class-agnostic manner introduced in Sec. 3.1.

**Details for R-CNN style experiments:** We provide more details here for the R-CNN style experiments: CLIP on cropped regions in Sec. 4.2 and ViLD-text+CLIP in Sec. 4.3. 1) Generalized object proposal: We use the standard Mask R-CNN R50-FPN model. To report mask AP and compare with other methods, we treat the second-stage refined boxes as proposals and use the corresponding masks. We apply a class-agnostic NMS with 0.9 threshold, and output a maximum of 1000 proposals. The objectness score is one minus the background score. 2) Open-vocabulary classification on cropped regions: After obtaining CLIP confidence scores for the 1000 proposals, we apply a class-specific NMS with a threshold of 0.6, and output the top 300 detections as the final results.

**Additional details for ViLD variants:** Different from the R-CNN style experiments, for all ViLD variants (Sec. 3.3, Sec. 3.4), we use the standard two-stage Mask R-CNN with the class-agnostic localization modules introduced in Sec. 3.1. Both the $M$ offline proposals and $N$ online proposals are obtained from the first-stage RPN (Ren et al., 2015). In general, the R-CNN style methods and ViLD variants share the same concept of class-agnostic object proposals. We use the second-stage outputs in R-CNN style experiments only because we want to obtain the Mask AP, the main metric, to compare with other methods. For ViLD variants, we remove the unnecessary complexities and show that using a simple one-stage RPN works well.

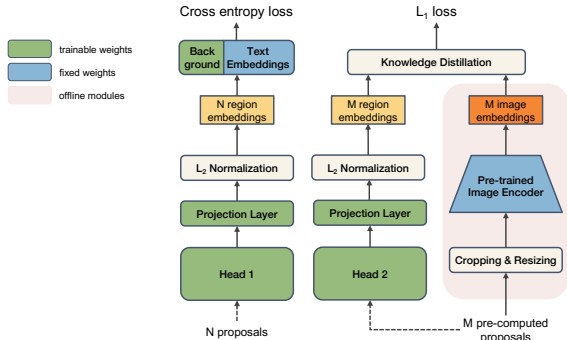

Figure 13: **Model architecture and training objectives for ViLD-ensemble.** The learning objectives are similar to ViLD. Different from ViLD, we use two separate heads of identical architecture in order to reduce the competition between ViLD-text and ViLD-image objetvies. During inference, the results from the two heads are ensembled as described in Sec. 3.4. Please refer to Fig. 3 for comparison with other ViLD variants.

**Architecture for open-vocabulary image classification models:** Popular open-vocabulary image classification models (Radford et al., 2021; Jia et al., 2021) perform contrastive pre-training on a large number of image-text pairs. Given a batch of paired images and texts, the model learns to maximize the cosine similarity between the embeddings of the corresponding image and text pairs, while minimizing the cosine similarity between other pairs. Specifically, for CLIP (Radford et al., 2021), we use the version where the image encoder adopts the Vision Transformer (Dosovitskiy et al., 2020) architecture and the text encoder is a Transformer (Vaswani et al., 2017). For ALIGN (Jia et al., 2021), its image encoder is an EfficientNet (Tan & Le, 2019) and its text encoder is a BERT (Devlin et al., 2019).

**Details for ViLD with stronger teacher models:** In both experiments with CLIP ViT-L/14 and ALIGN, we use EfficientNet-b7 as the backbone and ViLD-ensemble for better performance. We also crop the RoI features from only FPN level $P_3$ in the feature pyramid. The large-scale jittering range is reduced to [0.5, 2.0]. For CLIP ViT-L/14, since its image/text embeddings have 768 dimensions, we increase the FC dimension of the Faster R-CNN heads to 1,024, and the FPN dimension to 512. For ViLD w/ ALIGN, we use the ALIGN model with an EfficientNet-l2 image encoder and a BERT-large text encoder as the teacher model. We modify several places in the Mask R-CNN architecture to better distill the knowledge from the teacher. We equip the ViLD-image head in ViLD-ensemble with the MBConvBlocks in EfficientNet. Since the MBConvBlocks are fully-convolutional, we apply a global average pooling to obtain the image embeddings, following the teacher. The ViLD-text head keeps the same Faster R-CNN head architecture as in Mask R-CNN. Since ALIGN image/text embeddings have 1,376 dimensions (2.7× CLIP embedding dimension), we increase the number of units in the fully connected layers of the ViLD-text head to 2,048, and the FPN dimension to 1,024.

**Text prompts:** Since the open-vocabulary classification model is trained on full sentences, we feed the category names into a prompt template first, and use an ensemble of various prompts. Following Radford et al. (2021), we curate a list of 63 prompt templates. We specially include several prompts containing the phrase "in the scene" to better suit object detection, *e.g.*, "There is {article} {category} in the scene".

Our list of prompt templates is shown below:

```
'There is {article} {category} in the scene.'
'There is the {category} in the scene.'
'a photo of {article} {category} in the scene.'
'a photo of the {category} in the scene.'
'a photo of one {category} in the scene.'
'itap of {article} {category}.'
```

```
'itap of my {category}.'
'itap of the {category}.'
'a photo of {article} {category}.'
'a photo of my {category}.'
'a photo of the {category}.'
'a photo of one {category}.'
'a photo of many {category}.'
'a good photo of {article} {category}.'
'a good photo of the {category}.'
'a bad photo of {article} {category}.'
'a bad photo of the {category}.'
'a photo of a nice {category}.'
'a photo of the nice {category}.'
'a photo of a cool {category}.'
'a photo of the cool {category}.'
'a photo of a weird {category}.'
'a photo of the weird {category}.'
'a photo of a small {category}.'
'a photo of the small {category}.'
'a photo of a large {category}.'
'a photo of the large {category}.'
'a photo of a clean {category}.'
'a photo of the clean {category}.'
'a photo of a dirty {category}.'
'a photo of the dirty {category}.'
'a bright photo of {article} {category}.'
'a bright photo of the {category}.'
'a dark photo of {article} {category}.'
'a dark photo of the {category}.'
'a photo of a hard to see {category}.'
'a photo of the hard to see {category}.'
'a low resolution photo of {article} {category}.'
'a low resolution photo of the {category}.'
'a cropped photo of {article} {category}.'
'a cropped photo of the {category}.'
'a close-up photo of {article} {category}.'
'a close-up photo of the {category}.'
'a jpeg corrupted photo of {article} {category}.'
'a jpeg corrupted photo of the {category}.'
'a blurry photo of {article} {category}.'
'a blurry photo of the {category}.'
'a pixelated photo of {article} {category}.'
'a pixelated photo of the {category}.'
'a black and white photo of the {category}.'
'a black and white photo of {article} {category}.'
'a plastic {category}.'
'the plastic {category}.'
'a toy {category}.'
'the toy {category}.'
'a plushie {category}.'
'the plushie {category}.'
'a cartoon {category}.'
'the cartoon {category}.'
'an embroidered {category}.'
'the embroidered {category}.'
'a painting of the {category}.'
'a painting of a {category}.'
```

