# OpenReview forum: "Open-vocabulary Object Detection via Vision and Language Knowledge Distillation"
_ICLR.cc/2022/Conference — ICLR 2022 Poster_

### Official Review · Reviewer_LJFG · 2021-11-02

**Correctness:** 4
**Technical Novelty And Significance:** 2
**Empirical Novelty And Significance:** 4
**Recommendation:** 8
**Confidence:** 5

**Main Review:**

Strengths:

1. The paper studies an important research area with many potential applications. It is a step towards more scalable and efficient training without a massive amount of supervision.
2. The quantitative results are impressive, and provide valuable insights. Every row of Table 2 and Table 3 conveys a novel and interesting insight. Especially, the authors show that self-supervised embedding models trained on general-purpose image-caption data have the potential to outperform fully supervised models on real-world, long-tailed detection tasks.
3. The proposed method can also be used for open-vocabulary instance segmentation, which is novel and may inspire future work in this area.
4. Unlike prior work that relies on specialized pretraining, the proposed method decouples pretraining and training by using two separate models. This results in flexibility, because for instance, a large and complex model can be used for pretraining, while a fast, lightweight model can be used for distillation.

Weaknesses:
1. The proposed method has limited technical novelty. the authors distill an existing classification model on an existing detection model via an existing distillation method. Note that despite limited technical novelty, the empirical contribution is significant.

2. The paper is at some parts vague and skips some details. For instance:

    * The architecture of the text encoder is not explained anywhere.

    * In Section 3.1, it seems the output of the second stage of Mask R-CNN is used as proposal, instead of the output of the first stage (RPN), which is normally called proposal. This is confusing because the output of the second stage is also the detected objects.

    * There is an L2 normalization in Figure 3, which is not elaborated in the paper, and seems to be redundant as the authors use cosine similarity, which already normalizes the embeddings.

    * The authors do not mention how many base and novel classes they have in the LVIS experiments.

3. In VILD-text, pushing background proposals into a single point may cause them to collapse and not correctly map to novel class embeddings at test time. How do the authors address this problem?

4. The authors use most of the categories as base classes, and particularly choose categories that have frequent annotations. It is not clear how the model would perform with a smaller amount of supervised data (both fewer categories and fewer samples per category), and how smaller data affects each component (i.e. recognition and proposal)

5. The authors mention the CLIP baseline is slow, but do not provide quantitative speed metrics.



**Summary Of The Paper:**

The authors propose a new method for training object detection models that can generalize to novel object categories without bounding box annotations. Their key idea is to use an existing vision-language joint embedding model (teacher), and distill its knowledge into a typical object detection model (student). This way, the detection model learns to project each object proposal into an embedding space, where it can be compared to the text embedding of various object categories, and the closest one can be chosen. The teacher model is an existing work (CLIP), which is trained on millions of image-caption pairs, and hence relies on weak supervision. The proposed method achieves impressive results on novel object detection, outperforming not only the best open-vocabulary detection method, but also fully supervised detectors.

**Summary Of The Review:**

The paper presents a new solution for an important, yet unexplored, research problem. It achieves impressive SOTA performance and provides valuable empirical insights. Hence, I believe this paper would significantly contribute to the field.

---

> ### Author Response · Authors · 2021-11-23
> **Thanks for your review, we thoroughly address your concerns**
>
> Thank you for the valuable feedback! We address your concerns below.
>
> **Technical novelty and contribution**:
> 1. While knowledge distillation is a widely used technique, no one has ever explored thoroughly on how to distill an open-vocabulary classification model into a detector. There are many design choices to consider and we believe our work could provide insights in this direction.
> 2. We employ a widely-used object detector, Mask R-CNN, to demonstrate the generality of our method. As for the teacher model, as you mentioned, decoupling the pretraining makes our method work with off-the-shelf open-vocabulary image classification models, which is very **energy-efficient**. We experimented with two popular teacher models, CLIP and ALIGN, with very different network architectures (ViT and EfficientNet), to demonstrate the effectiveness of our method.
> 3. How to expand the vocabulary for object detection is a long-lasting open problem in computer vision. In the past, researchers have gravitated towards collecting larger and larger annotated detection datasets, from PASCAL VOC to COCO and recently to LVIS. This effort is now facing great challenges when scaling further due to the long-tailed nature of object categories. Our proposed method offers an alternative solution to this important open problem, by distilling from open-vocabulary image classification models pre-trained on weakly supervised image-text pairs.
>
> **Text encoder**:
> The text encoder is the one used in the pretrained open-vocabulary image classification model, i.e., CLIP/ALIGN (See Sec. 3 "Notations" and Fig. 2). And the text embeddings are extracted offline (Sec. 3.2), so $\mathcal{T}$ is not trained. We’ve updated Appendix D by adding a paragraph titled “Architecture for open-vocabulary image classification models”.
>
> **Object proposal**:
> In Sec. 3.1, we use the term "proposal" to describe a general concept (not the specific RPN proposed by Ren et al.): the class-agnostic localization of objects. For clarity, we have updated the title of Sec. 3.1 from “Object proposals for novel categories” to “Localization for novel objects”.
>
> **L2 norm**:
> The L2 normalization in Fig. 3 is a substep to compute the cosine similarity. In Fig. 3, the text embeddings are the counterpart of the learnable classifier in Fig. 3(a), and thus there will be no normalization for the region embeddings without plotting the L2 normalization (the text embeddings are normalized when we extract them offline). In Fig. 3(b), L2 normalization + classifier / dot product is equivalent to cosine similarity and this is also how we implement it.
>
> **Number of categories**:
> There are 337 novel categories and 866 base categories. We've updated the paper. Thanks!
>
> **Background**:
> Empirically, we find it works very well as demonstrated by our results. We suspect the possible reasons are: 1) among the background proposals, the number of real background regions is much larger than that of novel object proposals; 2) with distillation, the novel class text embeddings would still yield higher cosine similarities than the learned background embedding. We expect future works to propose explicit schemes to better handle the background.
>
> **Base/novel splits**:
> 1. Our setting is close to real-world scenarios, where it is easy to collect training examples for frequent and common categories.
> 2. Fewer categories: COCO has fewer categories than LVIS (48 vs 866 base categories). As shown in Table 4, we find the performance of ViLD-text on novel categories is much worse as fewer categories make the generalization difficult. In contrast, the performance of ViLD is robust, as the ViLD-image distillation is not affected by the number of base categories.
> 3. As for fewer samples, we do not study this scaling behavior. This is a very interesting research problem. With a small amount of supervised data, perhaps the initialization of the backbone is more important, or maybe new algorithms are needed. We leave it to future research.
>
> **Speed of CLIP on cropped regions**:
> We measured its runtime and the runtime is 630x of the runtime of ViLD (See the $^\dagger$ and caption in Table 3).

---

> > ### Comment · Reviewer_LJFG · 2021-11-29
> > **Feedback**
> >
> > The authors have addressed most of my questions, and I appreciate it. I look forward to seeing the paper at the conference.

---

### Official Review · Reviewer_Vqs3 · 2021-11-02

**Correctness:** 3
**Technical Novelty And Significance:** 3
**Empirical Novelty And Significance:** 4
**Recommendation:** 8
**Confidence:** 4

**Main Review:**

### Strengths

- The addressed problem is interesting and relevant for many practical applications
- The authors did a good job exploring different variants of how to leverage a pre-trained vision & language model (i.e., RPN+CLIP, ViLD-image/text/ensemble)
- The empirical results are very encouraging
- The proposed setting of using LVIS for open-vocabulary detection is good
- The paper (incl. appendix) provides several additional interesting experiments (transfer to other datasets, expanding of label space with attributes, etc.)


### Weaknesses

#### Clarity

- Region proposal networks (Sec 3.1 and 3.3): It is unclear to me how the authors implemented the region proposal network. Only after reading the whole paper, it seems that there are two different variants: The first one is a two-stage detector as described in Sec 3.1. The other is the standard RPN network used in Mask-RCNN, mentioned in Sec 3.3. This aspect is still unclear to me.  In any case, the two-stage objectness detector (Sec 3.1) needs more explanation. First, one should add a reference to the appendix already in Sec 3.1. But even then, some training details may be missing because in Mask-RCNN, the details of how to assign boxes to anchors/proposals is different region proposal vs. region classification networks.

- Related to the previous point, in Sec. 3.3, the authors mention the "two-stage detector", which was unclear to me. Does this refer to a standard Mask-RCNN with RPN or is this in reality a "three-stage" detector?

- Sec. 3 - Notations: It might be worthwhile spending one more sentence or a small figure on explaining the typical architecture of the pre-trained open-vocabulary image classification model.


#### Related work

- One could add a reference to the related problem of "open world object detection". Specifically, a recent work [A] introduces an "unknown-aware RPN", which might be relevant for this work as well.

- There have been a few recent works that try to expand the label space of object detectors by unifying the heterogeneous label spaces from multiple datasets into one model [B,C]. This seems like related work for the aspect of increasing vocabulary size.

#### Experiments

- Sec 4.5: It is not clear if the model used to compare on the COCO dataset uses FPN or not. If yes, I believe the comparison is unfair because the main competitor (Zareian et al.) seems to only use a ResNet-50-C4 backbone.


#### Minors and suggestions

- Sec 1: "exponentially more data is needed (Gupta et al., 2019), which makes it expensive to scale up detection vocabularies.". This statement is true for the images, but maybe not for the annotations, i.e., the human cost, because of the federated annotation technique that is used by LVIS (and also OpenImages >= v4). Please relate this statement to this type of dataset annotation.

- Please add a citation for the Average Recall (AR) metric, which is computed over multiple IoU thresholds. This may be unclear for readers not familiar with the details of object detection evaluation, and the numbers may then seem surprisingly low for a metric that measures recall.

- Sec 4.3: Typo: "... CLIP on cropped regions already outperforms supervised baselines ..."

- Implementation details: Are the input images first resized and then cropped to 1024x1024?

- Implementation details: What version of the CLIP model was used for distillation? CLIP provides pre-trained models for different image backbones, e.g., R-50 or ViT.

- Why was the R-50 backbone not initialized with the weights from the CLIP-R50 model? That could further add knowledge "transfer".

- Last paragraph in introduction: "... that uses additional tricks" is anecdotal. Please clarify. Also, I assume the challenge winners used full supervision for rare categories during training, which should be highlighted.

- Sec 4.6: The statement "... it is the first time we can directly transfer a trained detector to different datasets" may be inaccurate. The multi-dataset object detection paper [C] evaluates in a "zero-shot cross-dataset" setting (see Table 5). It is a different setting, but it's related, which should be discussed.

#### References mentioned above
- [A] Towards Open World Object Detection. Joseph et al. CVPR'21
- [B] Object Detection with a Unified Label Space from Multiple Datasets. Zhao et al. ECCV'20
- [C] Simple multi-dataset detection. Zhou et al. arXiv'21


**Summary Of The Paper:**

This paper addresses the task of open-vocabulary detection, where an object detector is trained from two forms of annotations: (a) bounding box annotations for a set of base categories and (b) open-vocabulary image-caption pairs.  This detector is then evaluated on a set of novel categories.  Compared to prior work, this paper tries to distill knowledge from strong pre-trained vision & language models (like CLIP) into an open-vocabulary detector. The authors propose multiple ways to do this (RCNN+CLIP, ViLD-text, ViLD-image, ViLD), including some ensemble variants (ViLD-ensemble). The paper also introduces a novel experimental setup based on the LVIS dataset (with >1200 categories) as compared to prior works that used COCO with only 80 categories.

**Summary Of The Review:**

Overall, I think this is a solid and interesting work. There are several aspects of the paper that need improvement, though, see comments above. I encourage the authors to consider those comments to further improve the manuscript.

---

> ### Author Response · Authors · 2021-11-23
> **Thanks for your review, we thoroughly address your concerns**
>
> Thank you for the helpful and constructive feedback! We address your concerns below.
>
> **Clarity (Region proposal networks):**
> 1. For clarity, we have updated the title of Sec. 3.1 from “Object proposals for novel categories” to “Localization for novel objects”.
> 2. In Sec. 3.1 we use the term “proposal” to describe a general concept (not the specific RPN proposed by Ren et al.): the class-agnostic localization of objects, which includes both the first-stage RPN and the second-stage box refinement and mask predictions (which can also be treated as proposals, as we will explain later).
> 3. The "two-stage detector" in Sec. 3.3 is referring to a standard Mask R-CNN with RPN. For all ViLD variants (Sec. 3.3, 3.4), we use a two-stage Mask R-CNN with the class-agnostic localization modules introduced in Sec. 3.1. Both the M offline proposals and the N proposals are obtained from the first stage RPN (the one proposed by Ren et al.).
> 4. For R-CNN style experiments (CLIP on cropped regions and ViLD-text+CLIP), we use the second-stage outputs as proposals, only because we want to compare them with other methods using Mask R-CNN on Mask APs. The architecture is the Mask R-CNN with class-agnostic localization modules, and the objectness score is simply 1 - background score. Nevertheless, these methods are not our main methods.
>
> **Clarity (architecture of the open-vocabulary image classification model):**
> Thanks for the suggestion! We’ve updated Appendix D by adding a paragraph titled “Architecture for open-vocabulary image classification models”.
>
> **Related work**:
> Thanks for pointing out those related papers! We've incorporated them into Sec. 2 of our updated paper.
>
> **Experiments (use FPN or not on COCO):**
> * Our model uses FPN. Besides the use of FPN, there are several other differences between ours and Zareian et al. that cannot be unified on COCO experiments, so we do not claim it to be an apple-to-apple comparison:
> 	* The pre-training sources (as listed in Table 4).
> 	* The model architecture of the pretrained models. We use a pretrained CLIP (ViT-B/32) and Zareian et al. use their proposed model (see Fig. 3 of Zareian et al.) to pretrain on COCO Captions.
> 	* How the pretrained models are used: Zareian et al. finetune from the pretrained model while we distill the knowledge from the pretrained model. Since they initialize the weights from the pretrained model, using a ResNet-50-C4 backbone can keep more information from pretraining, while using a FPN will introduce more randomly initialized weights. In contrast, our distillation method allows us to freely choose the detector's architecture that can be different from the teacher model (e.g., distill a ViT-B/32 into a ResNet-50).
> * As for the use of FPN, as shown in Detectron2’s model zoo (https://github.com/facebookresearch/detectron2/blob/main/MODEL_ZOO.md#coco-instance-segmentation-baselines-with-mask-r-cnn), R50-FPN (3x) introduces +1.2 box AP while its runtime is ~45% of that of R50-C4. This is the major reason why we choose to use FPN in our Mask R-CNN models. As shown in Table 4, the difference between ours and Zareian et al. is much more significant than 1.2 AP.
>
> **Minors and suggestions**:
> 1. To be more accurate, we've changed "exponentially" to "significantly". Thanks for the suggestion!
> 2. We added a citation for AR. We followed the AR metric used by COCO, which is averaged over 10 IoU thresholds of .50:.05:.95.
> 3. Typo: Thanks! Fixed.
> 4. Input size: The input images are first resized along the long edge, and then padded to 1024x1024.
> 5. As shown in the footnote of page 6, we use CLIP ViT-B/32.
> 6. Initialization: We use CLIP ViT-B/32 for its better performance. And the results demonstrate that the distillation method works even when the teacher model has very different architectures.
> 7. Challenge winner: Thanks! We added "fully-supervised" when describing their method. The challenge winner used at least 12 improvements (see table 1 and table 2 in [a]). Their major improvements include: 1) powerful data augmentations (Mosaic, instaboost etc.); 2) higher capacity backbone model; 3) self-training on OpenImages, etc.; 4) a complicated two-stage training, each employing specially designed loss functions; 5) multi-scale testing. We listed some of them in Sec 4.4 due to limited space and please refer to their tech report [d] for more details. We believe most of their tricks are **orthogonal** to our approach.
> 8. “It is the first time we can directly transfer …”: We added “using language” for clarity.
>
> [a] Tan et al. 1st Place Solution of LVIS Challenge 2020: A Good Box is not a Guarantee of a Good Mask. arXiv:2009.01559.

---

> > ### Comment · Reviewer_Vqs3 · 2021-11-29
> > **Acknowledging author feedback and further suggestions**
> >
> > Dear authors,
> >
> > thanks for your feedback on my review. It definitely helped me better understand more aspects of your work. I also hope that some of my suggestions will find its way into a camera-ready version. Here are a few final comments that you may want to consider for improving the manuscript:
> > - I think it would be good to include a clarifying sentence on the differences between proposals for RCNN-style and Two-stage-style detector experiments.
> > - Regarding the use of FPN. I understand why FPN was chosen over C4. I also understand that there are differences between Zareian et al. and your work. But I still believe one should try to strive for the fairest comparison possible, in a sense that, settings that can be controlled should be set equal. Obviously, your source of additional knowledge (CLIP and the corresponding dataset) and the way to use it (distillation) are different. But the network architecture, as you rightfully state, can be arbitrary for your method. Hence, I would find it better if an additional experiment with the R50-C4 architecture was conducted.
> > - By the way, I think it would be good to better highlight this flexibility of the choice of detector architecture as benefit over Zareian et al., in case you didn't do that already and I just missed it.

---

> > > ### Author Response · Authors · 2021-11-30
> > > **Thank you for the suggestions!**
> > >
> > > Thanks for your detailed suggestion and we will certainly consider them in the final version of our paper!

---

### Official Review · Reviewer_6g5Q · 2021-11-05

**Correctness:** 3
**Technical Novelty And Significance:** 3
**Empirical Novelty And Significance:** 3
**Recommendation:** 6
**Confidence:** 5

**Main Review:**

The paper presents strong results for some cases like rare classes in LVIS dataset. I would like to comment the authors on a well-written paper. It is mostly easy to understand. The authors have provided clear motivations and intuitions behind most of their design decisions. The diagrams and figures also assist in understanding the paper better.

However, there are still several problems with the paper which need to resolved before it can be considered ready for acceptance. The performance on non-rare classes and the complete test sets is significantly lower than prior works even though the performance on rare classes is slightly better in many cases. This raises questions about the utility of the proposed approach. Further, using models like CLIP which have probably already seen the rare classes muddies the definition of a "zero-shot" setting as used in the paper.

In particular, I have the following concerns/comments/questions/suggestions about the current version of the paper.

1. Do the authors know what classes the pre-trained image encoder (CLIP) is trained on? Has it seen any of the novel categories? If yes, then I would hesitate in calling it a zero-shot setting. I think the authors should just call it an open-vocabulary setting which can be weaker than a real zero-shot framework. Further, when comparing against prior zero-shot works, the authors need to clearly specify that the settings are slightly different because many of the prior works have ensured that no visual data from unseen classes has been used during pre-training or training.

2. It's not clear to be why we need N and M proposals separately? This is what I understand currently: The N proposals come from the RPN being trained along with the other trainable weights and might change from one epoch to the next. On the other hand, the M proposals are obtained from the RPN at the start of training and are then fixed i.e. they do not change throughout training. Is this correct? If yes, why not just use the N proposals in both cases? In section 3.4, the authors mention that using the same proposals causes contentions. I am not sure how or why. Further, if my interpretation mentioned above is incorrect, then how are the M "pre-computed proposals" obtained during training? And how are they obtained during inference? Similarly, for N.

3. In section 3.2, the authors mention that they ensemble 1x and 1.5x embeddings. Are there any emperical studies to show this helps? What if we just used 1x or 1.5x? What if we used 1x, 1.5x, and 2x? What if we used 0.5x and 1x? etc.

4. In section 3.4, the authors mention that "ViLD-image learns to approximate the predictions.....may improve performance". This seems very odd to me. If ViLD-image already approximates the predictions of the teacher model, why do we need to add the teacher model directly?

5. I am not sure I understand how ViLD-ensemble is different from ViLD. Is the diagram shown in Figure 3d ViLD or ViLD-ensemble? Can the authors please list all of the differences between ViLD and ViLD-ensemble?

6. In the caption of Table 3, the authors mention that the SOTA model uses additional tricks. What additional tricks? Are they not applicable to ViLD? What is the performance of ViLD using those tricks? The performance on all r, c, f, and overall is very low compared to SOTA. Why is this? And what is the utility of the proposed approach if this is the case?
 Further, I think not making the SOTA performance bold in Table 3 might be disingenuous. Some reasons I can think for not doing this are 1. if the SOTA performance achieved in a supervised setting, 2. the SOTA model uses the rare classes during training, 3. the SOTA model uses different/larger data for training. Are any of these true? If not, the authors should highlight the SOTA performance and discuss why the proposed approach does not meet SOTA and what can be done to achieve similar/better performance.

7. For table 4, what is the performance of supervised methods for COCO novel classes? The authors can do a similar comparison as done for LVIS. Further, the authors should clarify that the CLIP model in ViLD-image and the ViLD-text models have seen instances of $C_N$ during the large-scale pre-training which makes it a weaker zero-shot setting than prior works.



Typo:
On page 6, in the 5th line from the bottom, there is an extra "has".

**Summary Of The Paper:**

This paper aims to propose a novel approach for open-vocabulary or zero-shot object detection by leveraging models pre-trained on internet-scale data which might include many of the desired open-vocabulary categories. The proposed approach, called ViLD, uses the information learned by models like CLIP, and ALIGN to detect "novel" classes. The paper shows that the proposed approach can achieve better performance than supervised methods on rare classes in the LVIS dataset. However, the performance is much lower than state-of-the-art. Also, the performance on frequent classes in LVIS, and the overall performance is significantly lower than prior works. The authors also show that the proposed approach can achieve good performance improvements on the zero-shot COCO dataset - both for "unseen" and seen classes.

**Summary Of The Review:**

The paper proposes an interesting direction for open-vocabulary open detection. The authors have described most things about the proposed approach clearly and have written an easy-to-understand paper.

But the paper, in its current form, is not ready to be accepted. There are several concerns about the validity of the experimental set-up and comparisons against prior works. Further, there are several decisions taken by the authors which need clarification or more empirical evidence.



Edit:
After reading the other reviews and the authors' responses to all the reviews, I have increased my rating to 6.

---

> ### Author Response · Authors · 2021-11-22
> **Thanks for your review, we thoroughly address your concerns (Part 2 / 2)**
>
> **Q6:**
> 1. Please note that the comparison to the 2020 challenge winner [d] is not a fair comparison. We have it in our table for a reference to the best available fully-supervised model. The challenge winner used at least 12 improvements (see table 1 and table 2 in [d]). Their major improvements include: 1) powerful data augmentations (Mosaic, instaboost etc.); 2) higher capacity backbone model; 3) self-training on OpenImages; 4) a complicated two-stage training, each employing specially designed loss functions; 5) multi-scale testing. We listed some of them in Sec 4.4 due to limited space and please refer to their tech report [d] for more details. We believe most of their tricks are **orthogonal** to our approach. We follow the common practice in the community to not compare against methods used in a challenge. For example, recent methods [a, b, c] that achieved SOTA on LVIS fully-supervised training also do not compare their methods against the challenge winner [d].
> 2. All 3 reasons you mentioned are true. We've updated the paper to make it clear by adding the fact the challenge winner is fully-supervised.
> 3. The Supervised-RFS (Mahajan et al., 2018; Gupta et al., 2019) is the most **apple-to-apple** comparison to show the effectiveness of the proposed approach. Both models use the **same backbone architecture** and **same training settings**. We limit the differences to: 1) Supervised-RFS uses additional annotations from rare categories and information about class frequency; 2) our method uses a pre-trained teacher model (CLIP/ALIGN) for knowledge distillation. The table below shows our method outperforms Supervised-RFS by 0.1 overall AP.
> And when using a stronger teacher model and a stronger backbone (ViLD-ensemble w/ ALIGN), the performance of ViLD is greatly improved on all metrics.
>
> | Method                      |AP$_r$|AP$_c$|AP$_f$| AP  |
> |-----------------------------|------|------|------|-----|
> |EQL v2 (CVPR 2021) [a]       | 19.1 | 25.0 | 30.7 | 26.2|
> |RFS + RS Loss (ICCV 2021) [b]| 16.8 | 24.3 | 29.9 | 25.2|
> |LOCE (ICCV 2021) [c]         | 18.5 | 26.2 | 30.7 | 26.6|
> |Supervised-RFS               | 12.3 | 24.3 | 32.4 | 25.4|
> |ViLD-ensemble                | 16.6 | 24.6 | 30.3 | 25.5|
> |ViLD-ensemble w/ ALIGN (b7) | 26.3 | 27.2 | 32.9 | 29.3|
>
> 4. We also attach some most recent representative fully-supervised SOTA methods for long-tailed object detection evaluated on LVIS in the above table for comparison. These results support that Supervised-RFS is a strong baseline model and ViLD is comparable to these supervised methods trained with annotations from rare classes.
> 5. Utility of the proposed approach: In addition to strong performance, our object detection method is a more general modeling with open-vocabulary capability. This greatly improves the applicability of our method, e.g., interactive object detection with language (Fig. 1, Fig. 5-7), finetuning-free transfer (Table 5), which are beyond the utilities of fully-supervised methods.
>
>
> **Q7:**
> 1. The supervised performance of COCO is Novel AP=65.9 and Base AP=58.1.
> 2. Thanks for your suggestion. We've updated the training source to be more specific from “image-text pairs from Internet” to “image-text pairs from Internet (may contain $C_B \cup C_N$)” in Table 4.
>
> Thanks for pointing out the typo! We’ve revised it in our updated paper.
>
> [a] Tan et al. Equalization Loss v2: A New Gradient Balance Approach for Long-tailed Object Detection. CVPR 2021.
> [b] Oksuz et al. Rank & Sort Loss for Object Detection and Instance Segmentation. ICCV 2021.
> [c] Feng et al. Exploring Classification Equilibrium in Long-Tailed Object Detection. ICCV 2021.
> [d] Tan et al. 1st Place Solution of LVIS Challenge 2020: A Good Box is not a Guarantee of a Good Mask. arXiv:2009.01559.

---

> ### Author Response · Authors · 2021-11-22
> **Thanks for your review, we thoroughly address your concerns (Part 1 / 2)**
>
> Thanks for your review. ​​We carefully address your concerns below.
>
> **Q1:**
> 1. We would like to clarify our method is **open-vocabulary** object detection, not in the category of zero-shot learning. Our method is designed to address the challenge of data availability in object detection through using a teacher model (CLIP/ALIGN) that is pre-trained on a large-scale image-text dataset.
> 2. The original authors of CLIP don’t reveal the details of their image-text data, and we suspect that many categories in LVIS are seen when pre-training CLIP.
> 3. We listed the different training settings / sources in Table 4 when comparing with previous zero-shot detection methods.
>
>
> **Q2:**
> 1. In Sec. 3.3, we mentioned the offline M proposals are used to make the training more efficient, and also pointed to Sec. 3.2. on how we obtain these M proposals. The online N proposals can also be used for the training ViLD-image. However, in practice, this would require feeding thousands of proposals into the image encoder one by one (Sec. 3.2), which significantly increases the training time.
> 2. In Sec. 3.4, we mentioned the cross entropy loss of ViLD-text and the $\mathcal{L}_1$ distillation loss of ViLD-image have contentions because they perform two training objectives on the same set of region embeddings: one is from dataset annotations and the other is from the teacher model. We updated the text to "the same set of region embeddings" for clarity.
> 3. In Sec. 3.3, we mentioned the online N proposals are obtained from the first-stage RPN of the two-stage detector, in the same way as Mask R-CNN, which is also the case for inference. In Sec. 3.3 and Fig. 2, we showed that the M proposals (ViLD-image distillation) are only used for training.
>
>
> **Q3:**
> 	The results of using different crop sizes are listed in the table below. We evaluated ALIGN on the groundtruth bounding boxes. We see a significant boost when including more context (1x + 1.5x), and do not see further improvements when ensembling more sizes.
>
> |Crop sizes |Top1 Acc | Top 5 Acc |
> |-----------|---------|-----------|
> |1x         | 24.8    | 46.6      |
> |1x + 1.5x  | 29.4    | 54.4      |
> |1x + 1.25x + 1.5x | 28.9 | 53.6  |
>
>
> **Q4:**
> 	ViLD-image learns to approximate the teacher model through knowledge distillation, while a gap still exists between them (see the difference between “ViLD-image” and “CLIP on cropped regions” in Table 3). We find adding the teacher model helps the performance (“ViLD-text+CLIP” in Table 3) but its inference time is significantly longer and therefore this model is impractical. ViLD can be improved by using a stronger backbone and a stronger teacher model (EfficientNet-b7 with ALIGN in Table 3 and ResNet-152 in Table 8).
>
>
> **Q5:**
> 1. As described in the caption, Fig. 3(d) shows ViLD.
> 2. For ViLD-ensemble, as mentioned in Sec. 3.4, “we learn two sets of embeddings for ViLD-text (Eq. 2) and ViLD-image (Eq. 3) respectively, with two separate heads of identical architectures.” As a comparison, in ViLD, we use one set of embeddings with one head and therefore it may cause contentions. The contentions can be well addressed by ViLD-ensemble (see the comparison between “ViLD” and “ViLD-ensemble” in Table 3). We have updated the paper by adding equation references for clarity.

---

> > ### Comment · Reviewer_6g5Q · 2021-11-24
> > **Thanks**
> >
> > Dear authors,
> >
> > Thanks a lot for your responses to my comments/queries. I believe that most of my questions have been addressed. I am still not completely sure about the differences between ViLD and ViLD-ensemble. I would recommend that the authors add a small figure showing the difference, maybe as supplementary material.

---

> > > ### Author Response · Authors · 2021-11-29
> > > **Thank you!**
> > >
> > > Thank you for your suggestion!
> > > We will include a figure and more explanation for ViLD-ensemble in the appendix, for the final version.
> > >
> > > Below we plot the model architecture and training objectives for ViLD-ensemble for better understanding, using the same notation as in Fig. 3.
> > > Head1 and Head2 have identical architecture.
> > >
> > > During inference time, it uses the same architecture as in Fig. 2 bottom, except that the confidence scores are ensembled from two sets of region embeddings:
> > > We obtain K proposals from the RPN in the Mask R-CNN, and the K proposals are fed into both heads to obtain K region embeddings from head1 and another K region embeddings from head2.
> > > For each set of K region embeddings, we compute their cosine similarity with the text embeddings of the categories used for inference, and then apply a softmax to obtain the classification confidence scores for each set.
> > > Finally, scores from the two sets are ensembled as in Eq. 5 and Sec. 3.4.
> > >
> > > ```
> > >   Cross entropy loss                  L1 Loss
> > >           ▲                              ▲
> > >           │                              │
> > > ┌──────┬──┴───────────┐        ┌─────────┴───────────┐          ┌────────────────┐
> > > │Back  │    Text      │        │      Knowledge      │          │   M image      │
> > > │ground│ Embeddings   │        │    Distillation     ◄──────────┤  embeddings    │
> > > └──────┴──▲───────────┘        └─────────▲───────────┘          │                │
> > >           │                              │                      │(offline module │
> > >  ┌────────┴─────────┐           ┌────────┴─────────┐            │ in Fig 3, pink)│
> > >  │    N region      │           │    M region      │            │                │
> > >  │   embeddings     │           │   embeddings     │            └────────────────┘
> > >  └────────▲─────────┘           └────────▲─────────┘
> > >           │                              │
> > >  ┌────────┴─────────┐           ┌────────┴─────────┐
> > >  │L2 Normalization  │           │L2 Normalization  │
> > >  └────────▲─────────┘           └────────▲─────────┘
> > >           │                              │
> > >    ┌──────┴───────┐               ┌──────┴───────┐
> > >    │  Projection  │               │  Projection  │
> > >    └──────▲───────┘               └──────▲───────┘
> > >           │                              │
> > >    ┌──────┴───────┐               ┌──────┴───────┐
> > >    │              │               │              │
> > >    │    Head 1    │               │    Head 2    │
> > >    │              │               │              │
> > >    └──────▲───────┘               └──────▲───────┘
> > >           │                              │
> > >       N proposals                    M proposals
> > > ```

---

### Decision · Program_Chairs · 2022-01-20

**Decision:**

Accept (Poster)

**Comment:**

the aim of this work is to produce an open-vocabulary detector.  The approach is via knowledge distillation from existing large-scale V+L models, and the evaluation is based on novel classes with LVIS.  The reviewers were generally happy with the work (approach and results), but there were substantial points of clarification during discussion that need to be properly integrated into the final manuscript.